# The multi-tissue landscape of somatic mtDNA mutations indicates tissue-specific accumulation and removal in aging

**Monica Sanchez-Contreras[1†], Mariya T Sweetwyne[1*†], Kristine A Tsantilas[2], Jeremy A Whitson[1], Matthew D Campbell[3], Brenden F Kohrn[1], Hyeon Jeong Kim[4], Michael J Hipp[1], Jeanne Fredrickson[1], Megan M Nguyen[1], James B Hurley[2], David J Marcinek[3], Peter S Rabinovitch[1], Scott R Kennedy[1*]**

[1]Department of Laboratory Medicine and Pathology, University of Washington, Seattle, United States; [2]Department of Biochemistry, University of Washington, Seattle, United States; [3]Department of Radiology, University of Washington, Seattle, United States; [4]Department of Biology, University of Washington, Seattle, United States

**\*For correspondence:**
sandu@uw.edu (MTS);
scottrk@uw.edu (SRK)

†These authors contributed equally to this work

**Abstract** Accumulation of somatic mutations in the mitochondrial genome (mtDNA) has long been proposed as a possible mechanism of mitochondrial and tissue dysfunction that occurs during aging. A thorough characterization of age-associated mtDNA somatic mutations has been hampered by the limited ability to detect low-frequency mutations. Here, we used Duplex Sequencing on eight tissues of an aged mouse cohort to detect >89,000 independent somatic mtDNA mutations and show significant tissue-specific increases during aging across all tissues examined which did not correlate with mitochondrial content and tissue function. G→A/C→T substitutions, indicative of replication errors and/or cytidine deamination, were the predominant mutation type across all tissues and increased with age, whereas G→T/C→A substitutions, indicative of oxidative damage, were the second most common mutation type, but did not increase with age regardless of tissue. We also show that clonal expansions of mtDNA mutations with age is tissue- and mutation type-dependent. Unexpectedly, mutations associated with oxidative damage rarely formed clones in any tissue and were significantly reduced in the hearts and kidneys of aged mice treated at late age with elamipretide or nicotinamide mononucleotide. Thus, the lack of accumulation of oxidative damage-linked mutations with age suggests a life-long dynamic clearance of either the oxidative lesions or mtDNA genomes harboring oxidative damage.

## Editor's evaluation

Using the most accurate deep sequencing technology, duplex sequencing, these authors have detected over 89,000 independent somatic mtDNA mutations representing the largest catalog of somatic mtDNA point mutations during aging in a single study. The analysis of these mutations provides compelling evidence to dismiss the idea that reactive oxygen species are a driver of mtDNA mutagenesis, but suggests that ROS may be tissue dependent. These results should provide a fundamental understanding of mitochondrial DNA mutagenesis in aging that should appeal to a broad audience. The novel discovery is the significant presence of transversion mutations (C>A/G>T and C>G/G>C), which previously were assumed almost nonexistent. Moreover, the study finds that, unlike conventional mtDNA mutations, these transversions are not involved in clonal expansion and

do not accumulate with age; their relative presence varies very significantly between tissues and can be affected by drug interventions.

## Introduction

Genetic instability is a hallmark of aging (*López-Otín et al., 2013*). A mechanistic link between somatic mutations and age-related diseases such as cancer is clear, but their importance in other aging phenotypes, long hypothesized, is poorly understood (*Zhang and Vijg, 2018*). Recent surveys of non-diseased somatic tissues have shown that mutations are pervasive in the nuclear genome (nDNA), increase with age, and vary considerably between tissues (*Abascal et al., 2021*; *Li et al., 2021*). Additionally, these nDNA mutations commonly occur in cancer-associated genes, show evidence of selection and clonal expansion, and may play important roles in tissue regeneration and tumor suppression (*Colom et al., 2020*; *Martincorena et al., 2018*; *Martincorena et al., 2017*; *Martincorena et al., 2015*; *Zhu et al., 2019*). Collectively, these studies indicate a growing realization that somatic mutagenesis and clonal dynamics are likely an important determinant of human health during aging. While the accumulation of somatic mutations in the mitochondrial genome (mtDNA) with age has long been documented, the specific nature of their occurrence, and the consequences for aging, have remained unclear (reviewed in *Sanchez-Contreras and Kennedy, 2022*).

In vertebrates, mtDNA is a maternally inherited ~16–17 kb circular DNA molecule encoding 37 genes: 13 essential polypeptides of the electron transport chain (ETC), two ribosomal RNA genes, and 22 tRNAs. Mitochondria are involved in a broad range of crucial processes, including ATP generation via oxidative phosphorylation (OXPHOS), calcium homeostasis, iron-sulfur cluster biogenesis, regulation of apoptosis, and the biosynthesis of a wide variety of small molecules (*Kowaltowski, 2000*). These processes rely on mitochondria such that disruption of the genetic information encoded in mtDNA by mutation leads to dysfunction of these important processes and subsequently induces disease (*Wallace, 1999*). Unlike nDNA, mtDNA replication is largely independent of the cell cycle. The higher level of mtDNA replication, the absence of several cellular DNA repair pathways, and the lack of protection from histones results in mtDNA mutation rates ~100–1000× higher than that of nDNA (*Khrapko et al., 1997*; *Marcelino and Thilly, 1999*). Moreover, due to the coding density of mtDNA being higher than nDNA (~91% vs. ~1%), the probability that a mutation disrupts protein function is greater.

Observational studies have shown that the genetic instability of mtDNA in somatic cells is a fundamental phenotype of aging and may be involved in the pathogenesis of several diseases (reviewed in *Larsson, 2010*). Collectively, studies examining endogenous mtDNA mutations have shown low levels of G→T/C→A mutations and a preponderance of G→A/C→T and T→C/A→G transitions. This has been interpreted as being contrary to free radical theories of aging by suggesting that reactive oxygen species (ROS) are not the primary driver of mutagenesis in mtDNA (*Arbeithuber et al., 2020*; *Ju et al., 2014*; *Kennedy et al., 2013*; *Williams et al., 2013*; *Zheng et al., 2006*). Other notable patterns include an over-abundance of mutations in the mitochondrial Control Region (mCR), an unusual strand bias, a mutational gradient in transition mutations, and a unique trinucleotide mutational signature (*Ju et al., 2014*; *Kennedy et al., 2013*; *Sanchez-Contreras et al., 2021*; *Wei et al., 2019*). However, while the presence of somatic mtDNA mutations is well documented, a clear causative role in aging remains controversial (reviewed in *Sanchez-Contreras and Kennedy, 2022*).

One reason for this controversy stems from a poor understanding of when, where, and how somatic mtDNA mutations arise during the normal aging process. Most conclusions regarding the accumulation of mtDNA mutations during aging are based on a limited number of experimental models and tissue types, with data largely focused on brain and muscle due to their perceived sensitivity to mitochondrial dysfunction. Only a small number of pan-tissue surveys have been performed (*Li et al., 2021*; *Ma et al., 2018*; *Samuels et al., 2013*). Importantly, most of these prior studies made use of either 'clone and sequence' or conventional next-generation sequencing (NGS) to detect mutations. These approaches are technically limited in their ability to detect heteroplasmy below a variant allele fraction (VAF) of 1–2% (reviewed in *Salk et al., 2018*). The advent of ultrahigh-accuracy sequencing methods has shown that most heteroplasmies are present far below this analytical threshold (*Arbeithuber et al., 2020*; *Kennedy et al., 2013*). As such, determining the burden of somatic mtDNA

mutations in the context of normal aging lags well behind the efforts focused on nDNA. This is especially pertinent given the heterogeneous nature of tissue decline during aging.

Like the nDNA, somatic mutations in mtDNA have been proposed to be under selection (*Suen et al., 2010*). Cells have evolved several mitochondrial quality control pathways such as removal of damaged mitochondria by mitophagy and fusion/fission to maintain a healthy mitochondrial pool (*Youle and Narendra, 2011*). The formation and expression of deleterious mtDNA mutations is hypothesized to lead to a loss of mitochondrial membrane potential, mitochondrial dysfunction, and induction of mitophagy. This is a potential mechanism by which cells prevent mtDNA mutations from reaching a phenotypic threshold capable of altering cell homeostasis (*Rossignol et al., 2003*; *Rossignol et al., 1999*). Evidence for involvement of quality control machinery in removing somatic mtDNA mutations has been contradictory, with some indicating a clear role for mitophagy and fission/fusion, while other evidence indicates no effect (*Chen et al., 2010*; *Chen et al., 2015*; *Pickrell et al., 2015*; *Suen et al., 2010*). Thus, the role, if any, of the mitochondrial quality control pathways in targeting mtDNA mutations for removal remains unclear.

We and others have previously identified a mitochondrially targeted synthetic peptide, elamipretide (Elam; previously referred to as SS-31 and Bendavia), and the NADH precursor nicotinamide mononucleotide (NMN) as interventions that restore mitochondrial function and tissue homeostasis late in life (reviewed in *Yoshino et al., 2018*, and *Obi et al., 2022*). The specific mechanism(s) by which these two compounds ameliorate age-related mitochondrial dysfunction differ. Elam interacts directly with the inner mitochondrial membrane and membrane-associated proteins, stabilizing the mitochondrial ultrastructure and influencing cardiolipin-dependent protein interactions to improve ETC function leading to reduced oxidant production, preservation of membrane potential, and enhanced ATP production (*Campbell et al., 2019*; *Mitchell et al., 2020*; *Zhang et al., 2020*). In contrast, NMN is an NAD+ precursor molecule and acts by elevating NAD+ levels and providing additional substrate for mitochondrial ATP generation (*Guan et al., 2017*; *Martin et al., 2017*; *Yoshino et al., 2011*). Neither intervention is expected to directly alter mtDNA repair mechanisms. Therefore, we sought to test whether these interventions would reduce the prevalence of mtDNA mutations in aged tissues because of their demonstrated ability to improve mitochondrial structure and/or function.

We first addressed the relative dearth of high-accuracy data regarding age-related accumulation of mtDNA in mice across multiple tissue types. To that end, we used ultraaccurate Duplex Sequencing (Duplex-Seq) to identify organ-specific mtDNA mutation burden in heart, skeletal muscle, eye, kidney, liver, and brain in naturally aged mice (*Kennedy et al., 2014*; *Schmitt et al., 2012*). Intra-animal comparison allowed us to determine whether mtDNA mutation rates differ between organs while still accounting for inter-animal variation. Our findings point to the accumulation of somatic mtDNA mutations being a dynamic and highly tissue-specific process that can be modulated by one or more cellular pathways amenable to small molecule intervention.

## Results

To study the effects of aging on the accumulation of somatic mtDNA mutations across tissues, we used Duplex-Seq to obtain high-accuracy variant information across the entire mtDNA. We examined six different organ systems (heart, kidney, liver, skeletal muscle, brain, and eye) at two different ages (young = 4.5 months; N=5 and old = 26 months; N=6). These two age groups were chosen for their representation of the two extremes of the adult mouse lifespan while mitigating potential confounders related to development, sexual maturation, and survival selection at more advanced ages. These tissues vary on their dependence of mitochondria function and OXPHOS (*Fernández-Vizarra et al., 2011*). To minimize variation of cell type substructure within tissues between animals, care was taken to isolate similar regions of each organ, as described in the Materials and methods section. In total, we sequenced over 27.9 billion high-accuracy bases, corresponding to a grand mean post-consensus depth of 10,125× for all samples with reasonably uniform coverage among experimental groups and mice, with the exception of the Ori$_L$ (5160–5191) and several masked regions with high G/C content and/or repetitive sequences (*Figure 1—figure supplement 1* and *Figure 1—figure supplement 2*, *Figure 1—figure supplement 2*; *Supplementary file 1*—Table 5). We observed a combined total of 77,017 single-nucleotide variants (SNVs) and 12,031 small insertion/deletions (In/Dels) (≤15 bp in size) across all tissue, age, and intervention groups. Collectively, these data represent the largest collection of somatic mtDNA point mutations obtained in a single study to date and is second only to Lujan et

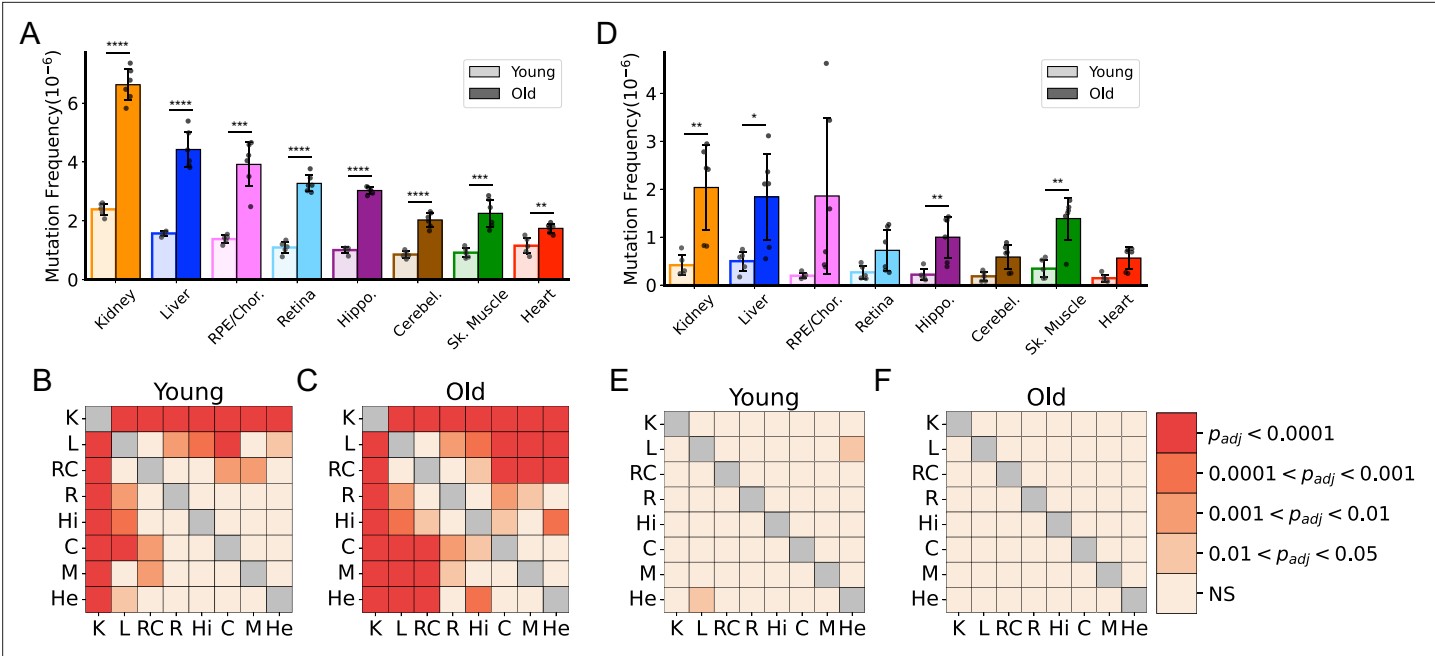

**Figure 1.** Frequency of somatic mitochondrial genome (mtDNA) mutations increase with age and is tissue-specific. (**A**) The frequency by which single-nucleotide variants (SNVs) were detected in all sequenced bases in either young (~5-months of age) or old (26-months of age) tissues arranged from highest to lowest SNV frequency in aged mice. (**B**) The frequency by which DNA insertions or deletions (In/Del) of any size are detected within all sequenced bases either young (~5-months of age) or old (26-months of age) tissues. For (**A**) and (**B**), significance between young and old within a tissue was determined by Welch's t-test. *0.01 < p < 0.05, **0.001 < p < 0.01, ***0.0001 < p < 0.001, ****p<0.0001; error bars = standard deviation of individual data points shown. (**C–D**) Heatmaps of one-way ANOVA with Tukey's HSD for significant differences of SNV frequencies between tissues, within either young (**C**) or old (**D**) age groups. (**E–F**) Heatmaps of one-way ANOVA with Tukey's HSD for significant differences of In/Del frequencies between tissues, within either young (**E**) or old (**F**) age groups.

The online version of this article includes the following figure supplement(s) for figure 1:

**Figure supplement 1.** Mean post-consensus 'duplex' depth for young (4.5 months) tissues.

**Figure supplement 2.** Mean post-consensus 'duplex' depth for old (26 months) tissues.

**Figure supplement 3.** Mitochondrial genome (mtDNA) copy shows no correlation with age, intervention, or mutation frequency.

**Figure supplement 4.** Blood does not significantly contribute to the differences in mutation frequency observed across tissues.

al. in terms of overall In/Del counts (*Lujan et al., 2012*). A summary of the data for each sample is reported in *Supplementary file 1*—Table 2 and *Supplementary file 2*.

## Frequency of somatic mtDNA mutations increase with age and is tissue-specific

To better understand the effects of aging on somatic mtDNA mutations across tissues, we determined the frequency of both SNVs and small In/Dels (≲15 bp) in aged mice. To minimize the contribution of mtDNA mutations that could be either maternally inherited or early clonal expansions established in development, we limited our analysis to mutations occurring at or below a VAF of 1%. In young mice, an initial comparison of the frequency of mtDNA SNVs revealed a mutation frequency on the order of ~1 × 10⁻⁶, with low variability between tissues (*Figure 1A*). Kidney and liver were notable exceptions, exhibiting significantly higher SNV frequencies compared to the other tissues in the young cohort (*Figure 1A and B*). With age, we observed significant increases in SNV frequency in all tissues we surveyed (*Figure 1A*). Moreover, mutation frequencies varied considerably between tissues in the aged cohort, with kidney having the highest SNV frequency (6.60±0.56 × 10⁻⁶) and heart having the lowest (1.74±0.16 × 10⁻⁶) (*Figure 1A and C*). The observed changes in frequency with age or tissue type did not correlate with differences in mtDNA copy number, as the mtDNA:nDNA ratio did not change with age (*Figure 1—figure supplement 3A, B*; *Supplementary file 1*—Table 3). In/Dels were approximately 10-fold less prevalent than SNVs in young mice, with a mean frequency of ~1.5 × 10⁻⁷,

and virtually no differences between tissues (*Figure 1D and E*). Like SNVs, In/Dels increased with age in most tissues we surveyed and did not correlate with copy number, but unlike SNVs, they did not significantly differ between tissue types, likely due to the high variability between samples (*Figure 1D and F*; *Figure 1—figure supplement 3C*).

Due to mutation burdens being tissue-specific, we considered whether these differences could be driven by variation in the contribution of mitochondrial mutations in leukocytes of circulating blood. To determine this, we analyzed Duplex-Seq in a small subset of tissues from aged mice perfused with PBS to remove the blood. Duplex-Seq of mtDNA from blood collected prior to the perfusion showed that in aged mice, the average frequency of SNV in blood was $3.05\pm0.15 \times 10^{-6}$, comparable to the frequency detected in aged hippocampus. Comparisons of perfused (no/low blood) to non-perfused tissues from liver, kidney, skeletal muscle, hippocampus, and cerebellum (the retina, retinal pigmented epithelium [RPE]/choroid, and heart were not sequenced), showed no significant difference in the frequency of SNV mutations (*Figure 1—figure supplement 4*). Thus, our mutation profiles are likely driven primarily by organ-specific cell types.

Although little is known about the kinetics of somatic mtDNA mutation accumulation during aging, they have been reported to increase exponentially during aging in mice (*Vermulst et al., 2007*). Both this current study and a prior study by Arbeithuber et al. report only two time points each (4.5 months vs. 26 months and 20 days vs. 10 months, respectively), making it impossible to confirm exponential increase in either study (*Arbeithuber et al., 2020*). However, the combination of our data with the previously published data by Arebiethuber et al. indicates a linear increase in overall mutation frequencies across the lifespan in the three tissue types common to both studies (brain, muscle, and liver). This indicates a likely constant 'clock-like' accumulation analogous to what is seen in the nDNA (*Abascal et al., 2021*; *Alexandrov et al., 2015*; *Arbeithuber et al., 2020*; *Figure 2*). Together, these data demonstrate that mtDNA mutations accumulate at tissue-specific rates during aging and indicate use of a single tissue source to draw broad organism level conclusions regarding the interaction between mtDNA mutations and aging is not scientifically supported.

## Mutation spectra of somatic mtDNA mutations demonstrate tissue-specific distribution of mutation types

Previous work by us and others indicates that somatic mtDNA mutations are strongly biased toward transitions (i.e. G→A/C→T and T→C/A→G), with low levels of transversions (*Ameur et al., 2011*; *Arbeithuber et al., 2020*; *Ju et al., 2014*; *Kennedy et al., 2013*; *Pickrell et al., 2015*; *Williams et al., 2013*). Moreover, due to their low prevalence, transversions associated with oxidative lesions (i.e. G→T/C→A and G→C/C→G) have been largely discounted as contributing to age-associated mtDNA mutagenesis (*Arbeithuber et al., 2020*; *Hoekstra et al., 2016*; *Itsara et al., 2014*; *Kauppila et al., 2018*; *Kennedy et al., 2013*; *Zheng et al., 2006*). However, these findings are based on a limited number of tissue types, specifically muscle and brain. Given the wide range of SNV frequencies and known metabolic activities of the tissues we assayed, we examined the mutational spectra for each tissue. Our data show that the overall bias toward G→A/C→T transitions remains broadly true for most tissues, but the extent of this bias varies considerably, with kidney and heart being the notable extremes (*Figure 3A*). In agreement with prior studies, a single mutation class, G→A/C→T, is the most abundant mutation type and accounts for more than 50% all mutations in most young tissues (*Figure 3—figure supplement 1*). In contrast, ROS-linked G→T/C→A and G→C/C→G mutations exhibited substantial variation in the level of mutations between tissues. In the central nervous system (CNS) tissues (hippocampus, cerebellum, retina), G→T/C→A and G→C/C→G, combined, accounted for an average of 23% of the total mutation burden (retina = 18%, hippocampus 18%, cerebellum 33%) (*Figure 3—figure supplement 1*). These data are consistent with prior Duplex-Seq-based studies that focused on neural tissues (*Arbeithuber et al., 2020*; *Hoekstra et al., 2016*; *Kennedy et al., 2013*). In contrast, skeletal muscle and heart in young animals have a relatively high frequency of ROS-linked mutations, with 43% and 66% of all mutations, respectively, resulting from these two types of mutations. This suggests that ROS is a greater source of mtDNA mutagenesis earlier in life and is tissue-dependent.

In comparison to the young tissues, mutation loads became more weighted toward transitions across the aged tissues we surveyed (*Figure 3B*). Significant differences between tissues within mutation classes also became more pronounced (*Figure 3B*, *heatmaps*). The fold-increase in most mutation types were remarkably uniform despite significant differences in SNV frequency between

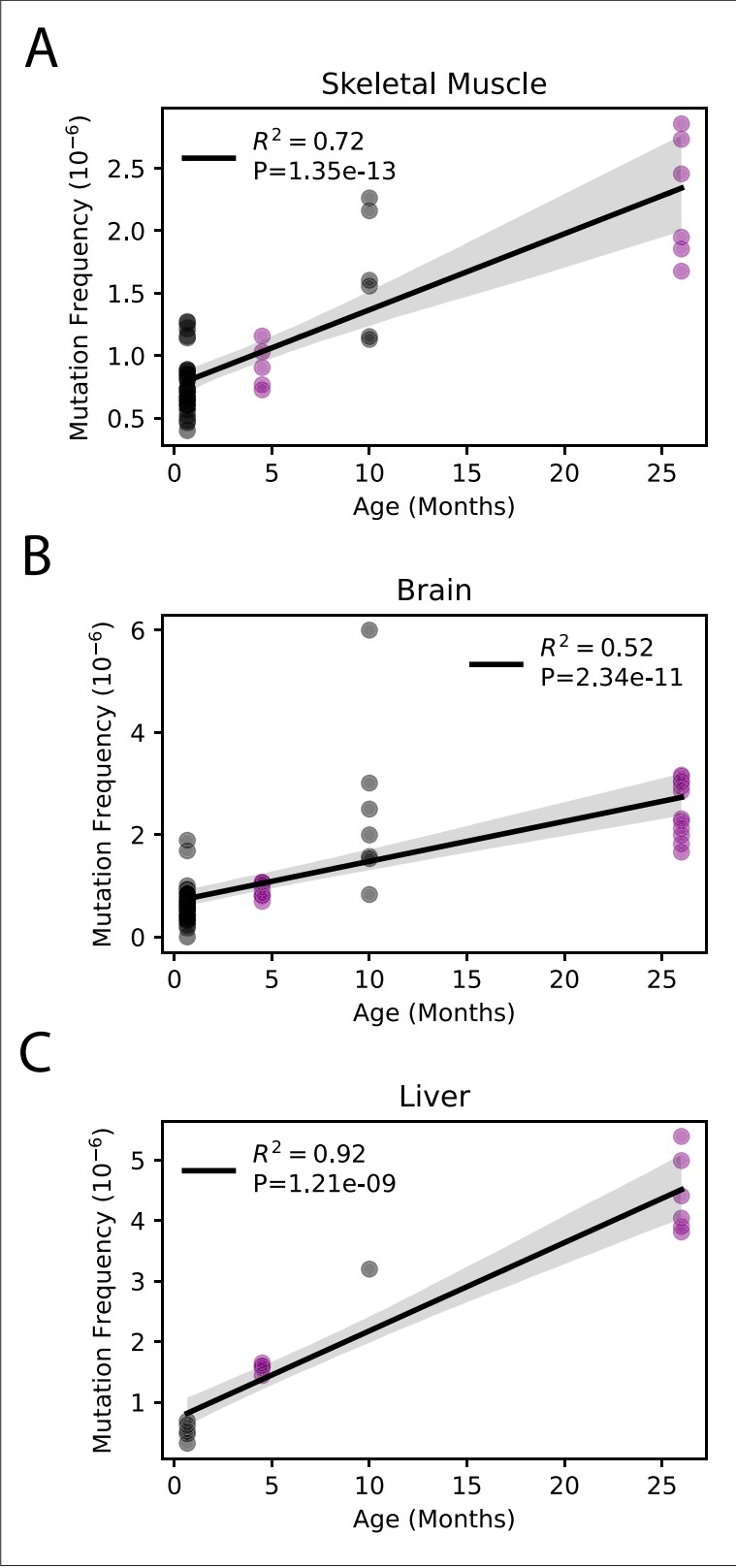

**Figure 2.** Somatic single-nucleotide variant (SNV) mutations increase linearly with age. Linear regression of total SNV mutation frequency vs. age in (**A**) skeletal muscle, (**B**) brain, and (**C**) liver. *Black* = data from Arbeithuber et al.; purple = data from this study; *shaded area* = 95% confidence interval of linear regression.

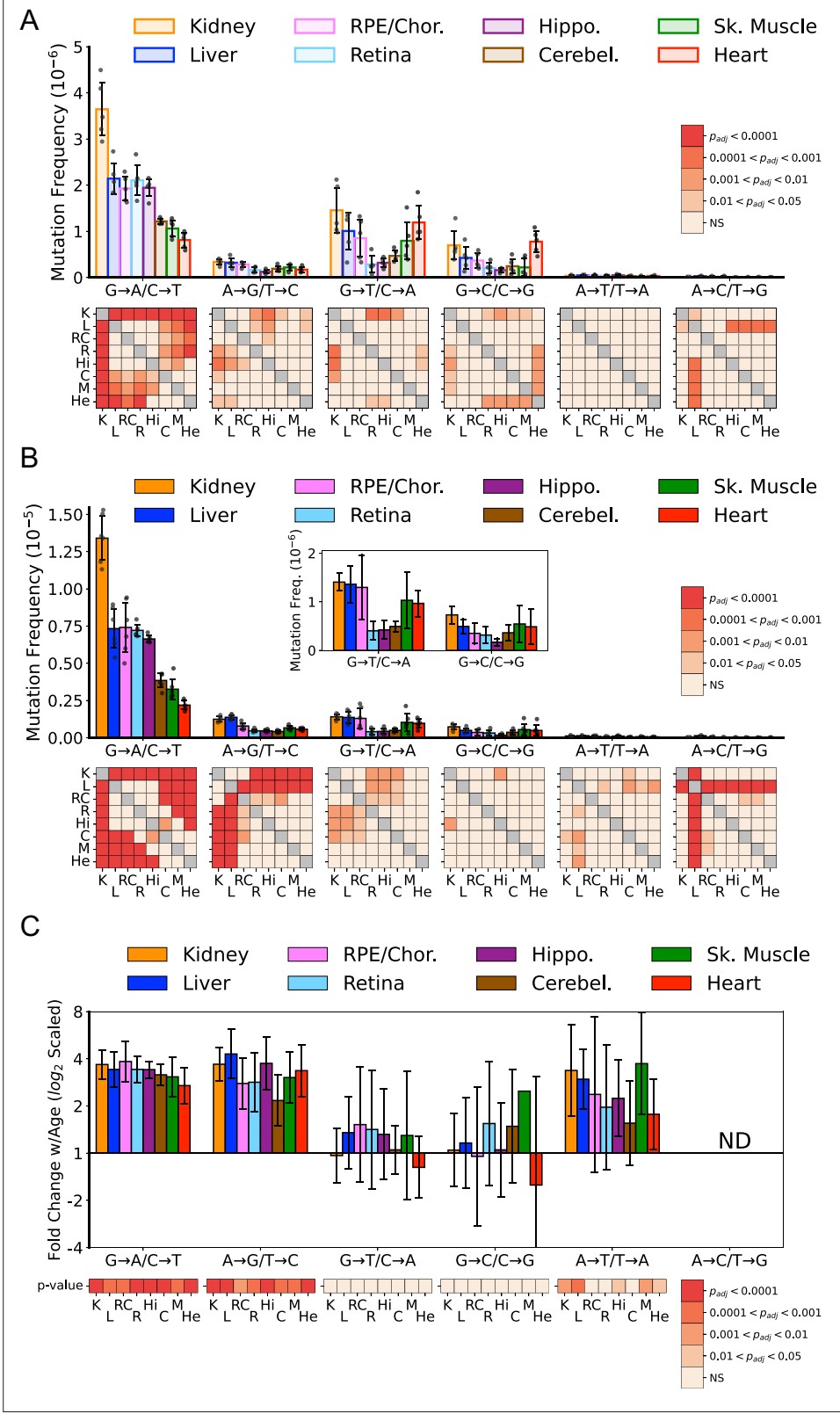

**Figure 3.** Mutation spectra of somatic mitochondrial genome (mtDNA) mutations demonstrate tissue-specific distribution of mutation types. (**A**) Single-nucleotide variant (SNV) frequency by mutation type for young (~5-months of age) tissues shows that replication/deamination-linked G→A/C→T mutations largely dictate overall SNV mutation burden and predominate in all young tissues except heart. Tissues of the central nervous system:

*Figure 3 continued on next page*

*Figure 3 continued*

eye retina, brain hippocampus, and brain cerebellum have the lowest frequencies of G→T/C→A and G→C/C→G transversions whereas they are highest in kidney and heart. Heatmaps show adjusted p-value from one-way ANOVA with Tukey's HSD for significant differences of SNV frequencies between young tissues within each mutation class. (**B**) SNV frequency by mutation type for old (26-month-old) tissues shows age-specific changes to mutation spectra. Heatmaps show one-way ANOVA with Tukey's HSD for significant differences of SNV frequencies between old tissues within each mutation class. (**C**) Fold-change of frequency from young to old age calculated for each tissue and spectra and shown with $\log_2$ scaling. Heatmap shows whether fold-change values of old relative to young mice are significantly different from fold-change 0 (no change). K=kidney; L=liver; RC = retinal pigmented epithelium (RPE)/choroid; R=retina; Hi = hippocampus; C=cerebellum; M=skeletal muscle; He = heart.

The online version of this article includes the following figure supplement(s) for figure 3:

**Figure supplement 1.** Relative proportion of different mutation types varies across tissues.

**Figure supplement 2.** The single-strand consensus G→T and C→A frequency does not vary across tissues in old mice.

**Figure supplement 3.** Treatment with FPG does not affect the G→T/C→T mutation frequency.

them (*Figure 3C*). Aging led to an average 3.2-fold increase of G→A/C→T and T→C/A→G transitions. Similarly, a significant 2.4-fold increase of T→A /A→T transversions was also observed (*Figure 3C*). A→C /T→G mutations were not evaluated due to their extreme paucity. In contrast to the other mutation types, G→T/C→A and G→C/C→G mutations did not significantly increase with age (*Figure 3C*).

The mtDNA has been documented to accumulate 8-oxo-dG in a tissue-specific manner (*Hamilton et al., 2001*). In addition, manipulations and high temperatures during library construction can further increase DNA damage (*Ahn and Lee, 2019*). It has been noted that DNA damage can, if present at sufficiently high levels, give rise to apparent G→T/C→A mutations in Duplex-Seq data, resulting from shearing induced DNA damage (*Abascal et al., 2021*; *Xiong et al., 2022*). We typically control for this class of artifacts by clipping the post-consensus read. However, artifacts have been documented to occur up to 30 cycles into DNA that is highly degraded (*Xiong et al., 2022*). To investigate the potential for DNA damage to explain the presence of G→T/C→A and G→C/C→G mutations, we performed two analyses. First, we examined the single-strand consensus sequence (SSCS) data that comprise the final high-accuracy duplex consensus sequence (DCS). Apparent variants in SSCS have sequencer-based errors removed and are comprised of both real DNA mutations and PCR-derived errors. Most of the PCR-derived errors are the result of base misincorporation across from DNA damage events, such as 8-oxo-dG and cytidine deamination to uracil (*Schmitt et al., 2012*). Our data show tissues significantly vary in the frequencies of G→T/C→A and G→C/C→G mutations. Therefore, if ROS-linked mutations were the result of fixation of DNA damage during PCR, then we would expect that the frequency of ROS-linked variants in the SSCS data should mirror the variability that is seen the final DCS data. Plotting the G→T and C→A SSCS frequency reveals that signal from ROS damage is uniform across all tissue types (*Figure 3—figure supplement 2*). Therefore, in order for false variants in the SSCS data to result in false DCS variants, the rate at which this occurred would need to be tissue-specific. Importantly, we randomized the library preparation and sequencing steps across tissues and ages to avoid batch effects, making it difficult to explain how tissue-level effects would occur. Second, we treated total DNA purified from aged skeletal muscle with formamidopyrimidine DNA glycosylase (Fpg) immediately after sonication. Fpg recognizes damaged purines, including 2,6-diamino-4-hydroxy-5-formamidopyrimidine and 8-oxo-dG, to generate an apurinic (AP) site that it then cleaves via its AP lyase activity, leaving a one nucleotide DNA gap. This effectively prevents the damaged DNA strand from PCR amplifying during library construction, thus preventing fragmented DNA from being able to form a final duplex consensus. Encouragingly, we observed no significant difference in our Fpg-treated and non-treated skeletal muscle samples, indicating that shearing-induced artifacts are unlikely to account for our observations (*Figure 3—figure supplement 3*). Taken together, our data suggest that ROS damage to the mtDNA is unlikely to explain our observations of tissue-related variability in G→T/C→A and G→C/C→G mutations.

## Clonal expansion of somatic mtDNA mutations is tissue-specific

Mutagenesis has been described as an irreversible process that results in increasing levels of mutations in a population over time, termed 'Muller's ratchet' (*Felsenstein, 1974*; *Muller, 1964*). Consequently, absent any compensatory mechanisms, mutations should increase during life. In the case of mtDNA, this should appear as an increase in the burden of apparent heteroplasmies (or clones) within a tissue over time. Importantly, because mtDNA replicates independently of nDNA, apparent heteroplasmies can increase during aging even in the absence of substantial cell proliferation. Moreover, mitochondria are subject to surveillance by mitophagy, which may affect the age-dependent mutational dynamics in tissue-specific ways (*Pickles et al., 2018*). Expansion of mtDNA mutations sufficient to warrant the term 'clonal' has been documented in human tissues, but the prevalence of this phenomenon remains poorly documented in mice (*Greaves et al., 2014*; *Greaves et al., 2010*; *Greaves et al., 2006*; *Nekhaeva et al., 2002*). The set of tissues we examined comprise a range of varying proliferative and replicative potentials, with heart, brain, and retina being limited, while many kidney and liver cell types proliferate throughout life. Therefore, we sought to determine the tissue-specific burden and dynamics of age-related clonal expansion of mtDNA mutations.

We defined a 'heteroplasmic clone' as a variant supported by three or more error-corrected reads and then calculated both the frequency and percentage of total mutations corresponding to these clones. We observed considerable tissue-specific variation in the effects of age on the presence of heteroplasmic clones, with all tissues exhibiting a significant increase in the frequency of total heteroplasmic clones with age (*Figure 4A and C*). Clones in all tissues were distributed relatively uniformly across the mtDNA coding region, but with a striking clustering of variants in the mCR (*Figure 4C*, *green region*), consistent with prior reports (*Arbeithuber et al., 2020*; *Kennedy et al., 2013*; *Sanchez-Contreras et al., 2021*).

The mutation composition of the clones varied between tissues. In the RPE/choroid, brain, skeletal muscle, and heart, the relative percentage of SNVs found as heteroplasmic clones did not change with age (~2–3% of total SNV). In contrast, kidney, liver, and retina exhibited a disproportionate increase in the number of clonally expanded variants with age. In old kidney, the percentage of SNVs detected as heteroplasmic clones increased to 10.4%, while clones in liver increased from ~1% of SNVs in young to 5.6% in old (*Figure 4B*). In retina, the percentage of SNVs detected as clones increased from 3.6% in young to 5.6% in old mice. In kidney and liver, the expansion of mtDNA mutations was pervasive across the genome and suggestive of a relationship to the high proliferative and regenerative capacity of these tissues. Retina, however, is a post-mitotic tissue and displayed a very different pattern, with the age-associated increase in clonality being attributed almost entirely to variants clustered in the mCR (*Figure 4C*).

Importantly, several of the tissues we examined are highly vascular and the hematological compartment has been documented to exhibit significant heteroplasmy and shifts in clonal expansions with age (*Lareau et al., 2019*). With the exception of kidney, which showed a modest effect, we observed no significant changes between the perfused and non-perfused samples, indicating that blood is unlikely a significant source of age-dependent changes across tissues (*Figure 4—figure supplement 1*). Collectively, these data suggest that the importance of mtDNA heteroplasmic clones in aging phenotypes is tissue-dependent. Very few studies have examined somatic mtDNA heteroplasmic clones in any tissue, and this remains an area for future work.

## Spectral analysis of clonal somatic mtDNA mutations suggests removal of ROS-linked mutations

Having established the tissue-specific profile of heteroplasmic clones, we reasoned that we could distinguish between mutations arising from a transient process earlier in life and an active clearance of mtDNA and/or whole mitochondria containing mutation types by examining which mutation types became more heteroplasmic with age. Specifically, an ongoing or early transient mutational process would be expected to result in expansion of a subset of variants across all mutation types. In contrast, evidence of active clearance would appear as either a lack of heteroplasmic clone expansion or a bias in the specific variants that underwent expansion. Because five of the eight tissues showed no significant change in the proportion of clonal SNV with aging, we expected that clones would be distributed across the spectra in a pattern similar to non-clonal mutations. Instead, we observed that the spectrum of heteroplasmic clones demonstrates a nearly complete suppression of heteroplasmic

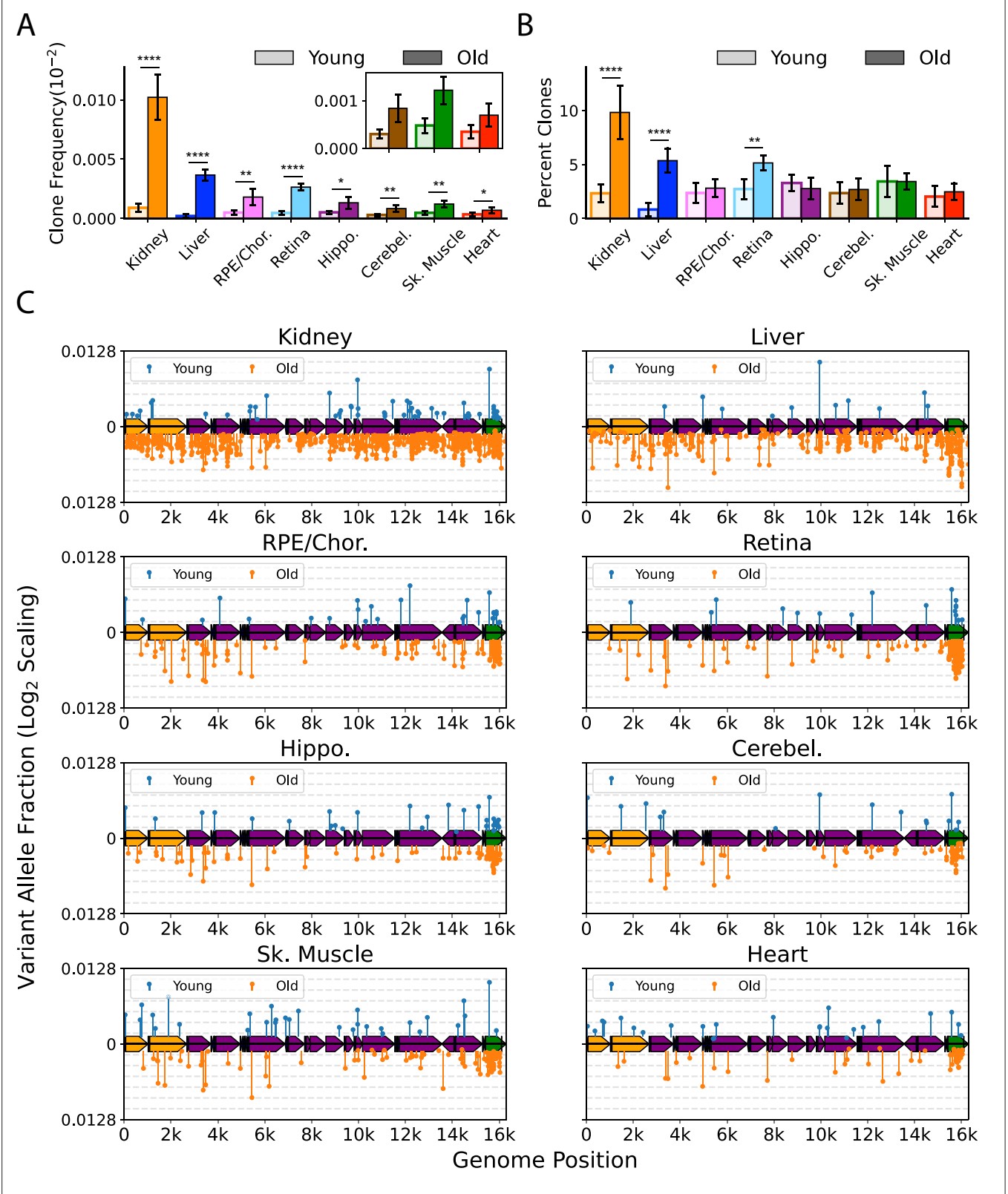

**Figure 4.** Clonal expansion of somatic mitochondrial genome (mtDNA) mutations is tissue-specific. (**A**) Frequency of mtDNA clones detected in each tissue shows an increase in detection of clones with age in all tissues. Note that y-axes are set for each tissue (N=5 for young; N=6 for old, error = ± standard deviation). (**B**) Percentage of total mutations found in heteroplasmic clones for each tissues shows that only kidney, liver, and retina have significant increases in relative 'clonality' with age. For (**A**) and (**B**), significance between young and old within a tissue was determined by Welch's t-test.

*Figure 4 continued on next page*

*Figure 4 continued*

*0.01 < p < 0.05, **0.001 < p < 0.01, ***0.0001 < p < 0.001, ****p<0.0001; error bars = standard deviation of individual data points shown. (**C**) Lollipop plots show the mtDNA genomic location of clonal hetroplasmic mutations in young (top row, blue markers, n=5) and old (bottom row, orange markers, n=6) for each tissue type. Orange = rRNA; dark blue = tRNA; purple = protein coding; green = Ori$_L$ or mitochondrial control region (mCR).

The online version of this article includes the following figure supplement(s) for figure 4:

**Figure supplement 1.** Mitochondrial genome (mtDNA) variant clones in blood are not a significant contributor to age-related to clonal expansions.

clones derived from G→T/C→A and G→C/C→G mutations in both young and old mice (*Figure 5A and B*). Remarkably, suppression of ROS-linked heteroplasmic clones was true even in the heart, which carried the highest combined G→T/C→A and G→C/C→G SNV mutation burden of any tissue, with 65% of total SNV in young and 34% of SNV in old mice (*Figure 3—figure supplement 1*). Thus, we asked whether this lack of G→T/C→A and G→C/C→G clones was a significant finding or merely a consequence of low sampling due to the relative paucity of heteroplasmic clones and the lower frequency of ROS-associated mutations. Under the assumption that heteroplasmic clones arise randomly, we tested whether our dataset of detected SNV clones differed from the expected number of clones based on the spectral distribution of non-clonal SNV mutations. To ensure that we had sufficient power, we combined SNV mutations and clones detected in all eight tissues of the six old mice for a total of 24,244 de novo SNVs and 1461 heteroplasmic clones. To account for differences in depth between samples, the spectral distribution of total SNV mutations was calculated for each tissue of each mouse. The expected contribution of clones in each tissue was then weighted based on the average percentage of 'clonality' measured in the aged dataset (*Figure 5B*). For example, more clones were expected to form in kidney and liver (10.4% and 6% clonality, respectively) than would be expected in brain or muscle tissues (~3% clonality). Using this method, our model predicted the detection of 1341 heteroplasmic clones in total for combined aged tissues, which is within 10% of the detected total of 1461.

We modeled the expected spectrum of these clones among the six mutation types using a Poisson process to model random sampling error due to the low abundance of clones. We compared the expected number to the observed spectrum and found that the clonal spectra differed significantly from the distribution of non-clonal SNVs (*Figure 5C and D*). G→T/C→A and C→G/G→C mutations were predicted to form ~146 and ~69 clones respectively, however, only eight G→T/C→A clones and two C→G/G→C clones were detected in the entire aged mouse dataset, corresponding to an 18- and 34-fold under-representation, respectively. Conversely, G→A/C→T and T→C/A→G transitions only deviated from the expected values by less than twofold (*Figure 5D*). Although we used a combined aged tissue dataset to ensure that we were not under sampling, this under/over-representation by mutation spectra was detected in every tissue type as shown by their observed spectral distribution (*Figure 5E*). Taken together, these results suggest that expansions of heteroplasmic clones in mtDNA unlikely arises as a random consequence of somatic mutation formation. The uneven distribution of clones relative to non-clonal SNVs suggests that the lack of G→T/C→A and G→C/C→G mutation accumulation with age does not reflect differences in the formation of these mutations, but rather is consistent with a steady-state level of ROS-linked mutations that is susceptible to a constant generation and removal.

## Late-life treatment with mitochondrially targeted interventions reduces ROS-linked mutations

Like in the germline, mtDNA mutations have been hypothesized to be selectively removed in somatic tissue through a mechanism involving a still poorly understood interaction between the unfolded protein response, mitophagy, and mitochondrial fission/fusion (*Chen et al., 2010*; *Gitschlag et al., 2016*; *Lin et al., 2016*; *Suen et al., 2010*). Therefore, we hypothesized that compounds known to improve function and/or ultrastructure in aged mitochondria would impact the burden of aging mtDNA mutations by shifting the steady state of ROS damage toward removal of damaged/dysfunctional mitochondria and their accompanying mtDNA. To this end we sequenced tissues from mice treated systemically for 8 weeks with either Elam or NMN in old mice that had accumulated somatic mtDNA mutations throughout their lives. Functionally, both Elam and NMN improve mitochondrial energetics and the mitochondrial network in aged mice across multiple aged tissues within an 8-week time frame

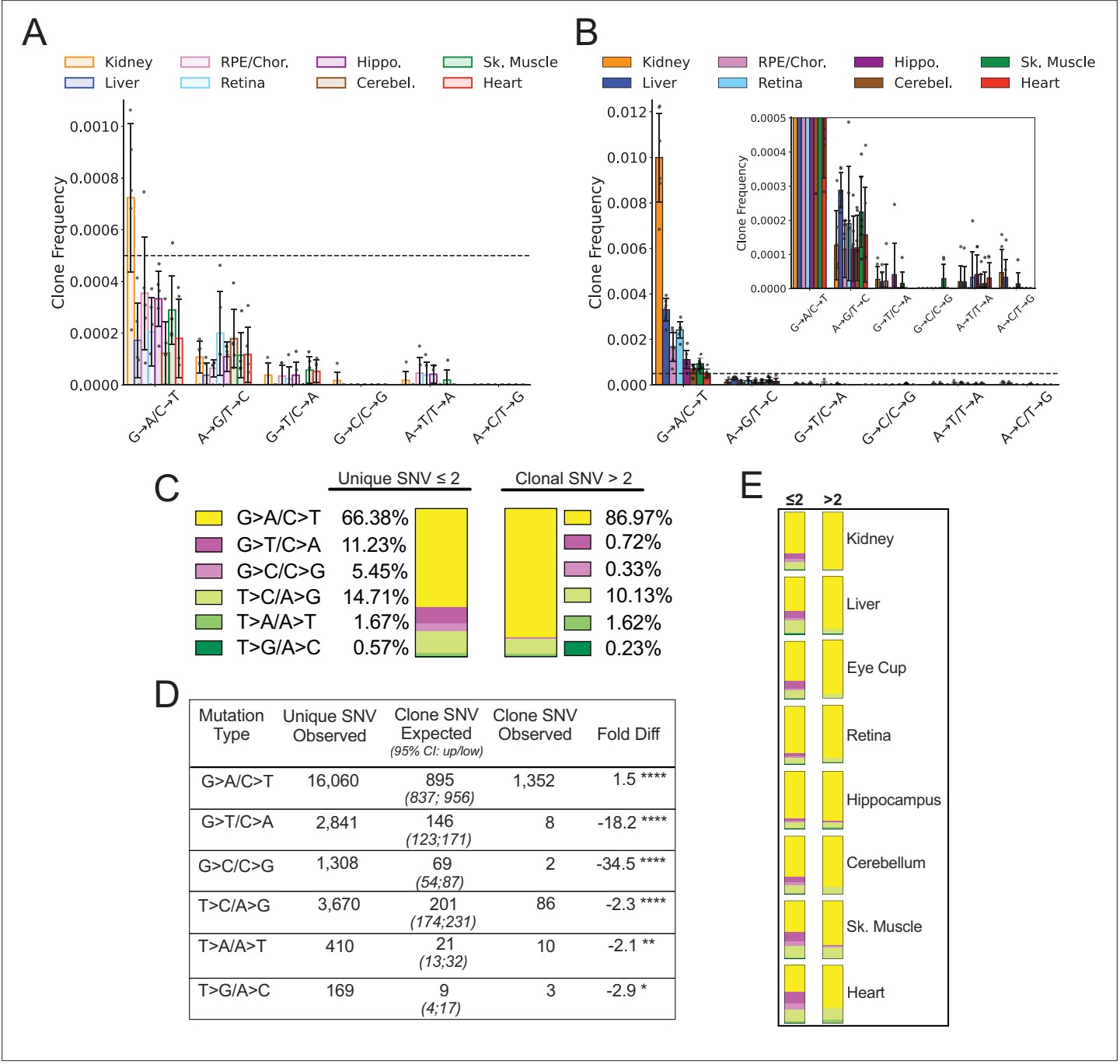

**Figure 5.** Spectral analysis of clonal somatic mitochondrial genome (mtDNA) mutations suggests removal of reactive oxygen species (ROS)-linked mutations. (**A**) Frequency of heteroplasmic clones in young mice shown as clone frequency for each mutation class and tissue. (**B**) Frequency of heteroplasmic clones in aged mice for each mutation class and tissue. Inset shows graph with adjusted axis to match young mice in (**A**) to better visualize lack of mutations in G→T/C→A and G→C/C→G mutation classes despite expansion of clonality with age. In both (**A**) and (**B**) the dotted line indicates frequency value of 0.005. (**C**) The combined distribution of mutation spectra for single-nucleotide variants (SNVs) for either unique mutations (detected two or less times) or clonal mutations (detected more than two times) for all aged tissues. (**D**) Table showing that, based on the mutation spectra of unique mutations, the observed number of SNV clones differs the number SNV clones expected if heteroplasmic clones arise randomly as a consequence of mutation burden. G→A/C→T and T→C/A→G clones are over-represented, while G→T/C→A and G→C/C→G clones are strongly under-represented based on Poisson sampling. 'Fold Diff' represents fold-change of observed clone values relative to expected. (**E**) Mutation spectra distributions for each aged tissue type mirrors the pattern of the combined aged samples with over/under representation of specific mutation types within observed clones. Mutation types are color coded as in (**C**).

The online version of this article includes the following source data for figure 5:

**Source data 1.** Spreadsheet containing the calculation of expected vs observed clone counts for *Figure 5C*.

(*Campbell et al., 2019*; *Chiao et al., 2020*; *Sweetwyne et al., 2017*; *Whitson et al., 2020*), albeit through different mechanisms. Specifically, Elam improves mitochondria structure and integrity of the inner mitochondrial membrane cristae (*Birk et al., 2013*; *Machiraju et al., 2019*), whereas NMN acts as a precursor molecule for NAD$^+$/NADP production (*Hong et al., 2020*). This allowed us to examine mice within such a narrow window of time that it was more likely to detect changes in mutation turnover/removal, rather than significant prevention of mutation accumulation during aging.

All treated samples were sequenced to a similar mean 'duplex' depth and detected comparable numbers of mutations as the controls (*Figure 1—figure supplement 2*; *Figure 6—figure supplement 1* and *Figure 1—figure supplement 2*; *Supplementary file 1*—Table 2). We did not observe a significant change in the overall mutation frequencies between aged mice and treated mice, regardless of tissue, indicating that these interventions are unlikely to indiscriminately affect mtDNA mutations (*Figure 6—figure supplement 3*). In support of this observation, the non-synonymous to synonymous ratio (dN/dS), which is a measure of positive or negative selection, shows no significant deviation from the expected ratio of one for any age, tissue, or intervention (*Figure 6—figure supplement 4*). However, consistent with our hypothesis, mice treated with Elam or NMN showed a trend for reduction of ROS-associated G→T/C→A and G→C/C→G mutations in heart, kidney, liver, and skeletal muscle. In heart, this reached significance for reduction of G→T/C→A mutations (control: $9.7\pm3.0 \times 10^{-7}$ vs. ELAM: $5.5\pm1.5 \times 10^{-7}$, $p_{adj}$ = 0.019; control: $9.7\pm3.0 \times 10^{-7}$ vs. NMN: $4.7\pm0.9 \times 10^{-7}$, $p_{adj}$ = 0.017; one-way ANOVA within each mutation class with Dunnett's test against saline control) (*Figure 6A*). In kidney, reduction of G→C/C→G mutations reached significance in NMN-treated animals (control: $7.3\pm2.0 \times 10^{-7}$ vs. NMN: $3.7\pm1.7 \times 10^{-7}$, $p_{adj}$ = 0.038; one-way ANOVA within each mutation class with Dunnett's test against saline control) (*Figure 6B*). Both liver and skeletal muscle trended toward significance (*Figure 6C and D*). No effect was seen in the remaining tissues (*Figure 6—figure supplement 5*). G→T/C→A and G→C/C→G mutations do not significantly increase between young and old age control animals, but are reduced in aged animals with these interventions, thus indicating that these changes are not simply due to the prevention of these mutation types during the treatment window. We interpret these findings to mean that ROS-linked mutations do occur throughout life, but that either mitochondria or the mtDNA harboring these mutations are more likely than other mutation classes to be eliminated via mechanisms of cellular organelle maintenance. Although we cannot discount that the lack of efficacy in some tissues may be due to tissue-specific differences in biodistribution of the two compounds tested, the efficacy of Elam and NMN to reduce ROS-linked mutations in these organs is consistent with other reports demonstrating functional improvements to aged tissues specifically in skeletal muscle, heart, and kidney for Elam (*Chiao et al., 2020*; *Sweetwyne et al., 2017*; *Whitson et al., 2020*) and heart, liver, and skeletal muscle for NMN (*Luo et al., 2022*; *Mills et al., 2016*; *Whitson et al., 2020*).

In contrast to the pattern seen in peripheral organs of treated mice, a reduction of ROS-linked mutations was not observed in tissues of the brain or eye (*Figure 6—figure supplement 5*). Both ELAM and NMN can cross the blood-brain barrier with ease and so are expected to be available to both brain and eye with systemic treatment. Our findings are consistent with the lower overall G→T/C→A and G→C/C→G mutation burden in these latter tissues and may indicate that their mtDNA is more protected from ROS damage with age than is the case in peripheral organs. When considered in the context of the data presented in *Figure 4*, which demonstrates a lack of clone formation for G→T/C→A and G→C/C→G mutations, the selective reduction of these same mutations in tissues with high oxidative mutation burden further supports the hypothesis that ROS-linked mtDNA mutation accumulation is regulated through life-long and specific removal of ROS-damaged mtDNA genomes.

## Discussion

The processes that drive and influence somatic mtDNA mutagenesis in aging and disease has proved to be nuanced and controversial (*Kauppila and Stewart, 2015*; *Sanchez-Contreras and Kennedy, 2022*; *Szczepanowska and Trifunovic, 2017*). Enhancement of accuracy by NGS and newer methods, such as Duplex-Seq, have begun to shed light on these processes. In this report, we took advantage of our continued improvement in the Duplex-Seq protocol to perform a multi-tissue survey of somatic mtDNA mutations in a naturally aging mouse cohort. The very high depths we achieved (depth grand mean: 10,125×) allowed us to detect hundreds to thousands of mutations per sample (variant count mean: 453, total: 77,017 SNVs and 12,031 In/Dels) across all mutation types, representing an ~8.3-fold

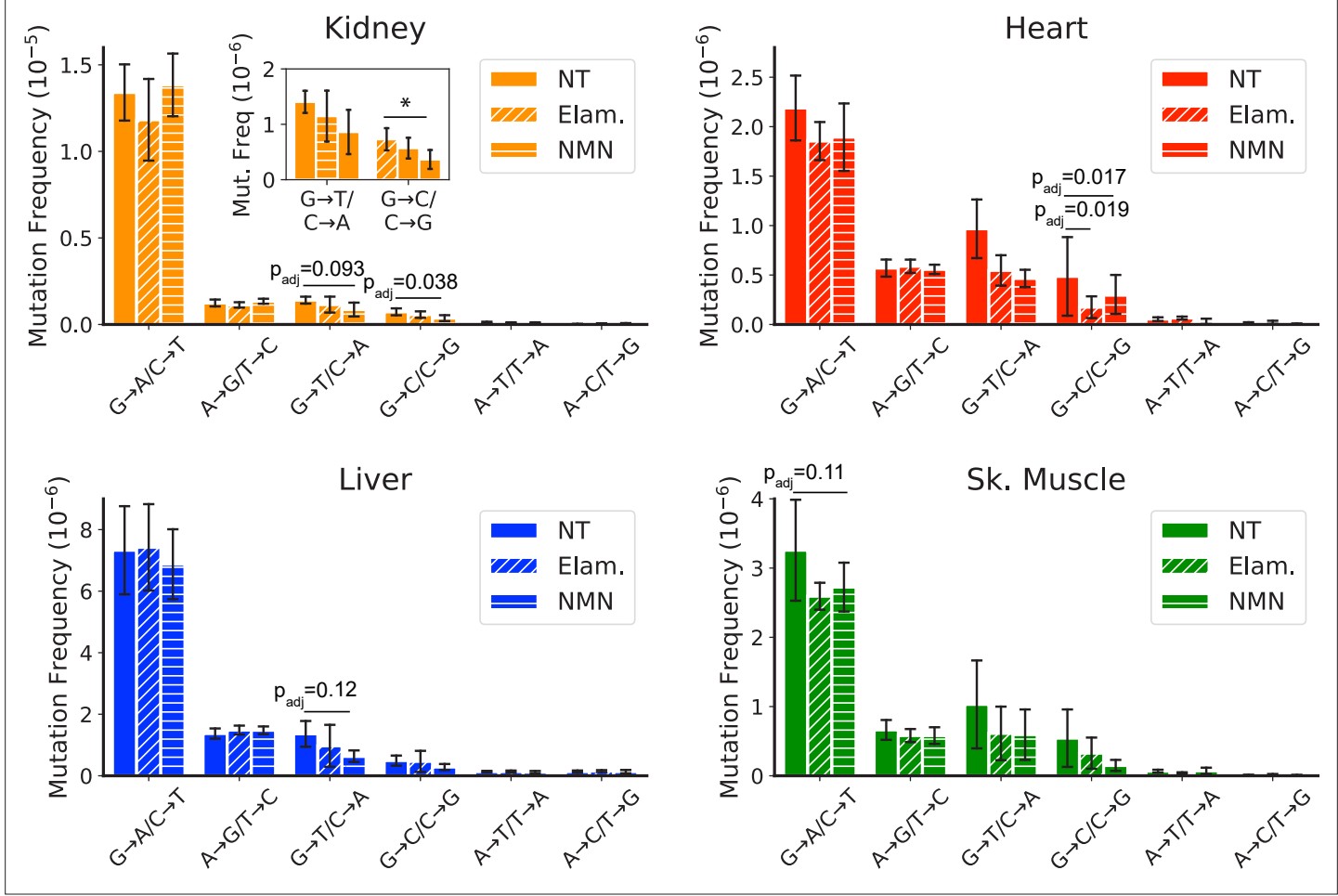

**Figure 6.** Late-life treatment with mitochondrially targeted interventions reduces somatic mitochondrial genome (mtDNA) mutation burden and is consistent with a mechanism of active reactive oxygen species (ROS)-linked mutation removal. Mutation spectra show that aged mice treated for 8 weeks with either elamipretide (Elam, diagonal striped bars) or nicotinamide mononucleotide (NMN, horizontal striped bars) have decreased frequency of mutations compared to no-treated (NT) controls specifically in mutation types linked to oxidative damage, G→T/C→A and G→C/C→G, specifically for (**A**) kidney and (**B**) heart and trending for (**C**) liver. (**D**) Muscle trends lower for Elam-treated mice in G→A/C→T mutations. Error bars = ± standard deviation. Statistics calculated by one-way ANOVA for each mutation class within a tissue and Dunnett's multiple comparison test compared to the untreated aged control group, significance=$p_{adj}$ <0.05 and trend=$p_{adj}$ <0.15.

The online version of this article includes the following figure supplement(s) for figure 6:

**Figure supplement 1.** Mean post-consensus 'duplex' depth for aged (26 months) elamipretide-treated tissues.

**Figure supplement 2.** Mean post-consensus 'duplex' depth for aged (26 months) nicotinamide mononucleotide-treated tissues.

**Figure supplement 3.** Elamipretide (Elam) and nicotinamide mononucleotide (NMN) do not affect the overall mitochondrial genome (mtDNA) mutation frequency.

**Figure supplement 4.** Per gene dN/dS ratio shows no apparent selection across age, tissues, and interventions.

**Figure supplement 5.** Late-life treatment with mitochondrially targeted interventions does not reduce somatic mitochondrial genome (mtDNA) mutation burden in retinal pigmented epithelium (RPE)/choroid (pink), retina (cyan), hippocampus (purple), or cerebellum (brown).

increase in depth and a 38.5-fold increase in mean mutation count per sample compared to the next largest currently published Duplex-Seq dataset for mouse mtDNA (depth grand mean: 1231×; variant count mean: 12.7, total: 2488) (**Arbeithuber et al., 2020**). This substantially expanded dataset allowed us to examine the types and classes of de novo mtDNA mutations with unprecedented sensitivity and, as a result, we observed unexpected patterns that would have been missed with a lower coverage of mutations.

The findings reported here broadly recapitulate those from smaller studies reporting that somatic mtDNA point mutations occur at a frequency on the order of $10^{-6}$, increase with age, and are biased

toward G→A/C→T transitions (*Ameur et al., 2011*; *Andreazza et al., 2019*; *Arbeithuber et al., 2020*; *Hoekstra et al., 2016*; *Kennedy et al., 2013*; *Samstag et al., 2018*; *Williams et al., 2013*). However, our expanded dataset found an unexpected level of variation between tissues in both overall mutation frequency and spectrum, indicating tissue-specific effects of aging on mtDNA mutation burden. While in young mouse tissues, we observed minimal variation in mtDNA SNV frequency with only kidney, liver, and RPE/choroid showing significantly increased levels compared to other tissues, with advancing age were distinct differences between all tissues apparent. Our data indicate that the accumulation of somatic mtDNA mutation is highly segmental in nature and therefore its impact on aging or disease risk may be tissue-dependent.

## The relevance of ROS-linked mutations in mtDNA and aging

In contrast to mutations arising from DNA polymerase γ (Pol-γ) error, ROS-linked mutations are unlikely to be specifically generated by the process of DNA replication itself. The studies that have examined mtDNA mutagenesis have noted a distinct lack of G→T/C→A and G→C/C→G transversions, suggesting that oxidative damage is not a major contributor to aging mtDNA mutagenesis (*Ameur et al., 2011*; *Andreazza et al., 2019*; *Arbeithuber et al., 2020*; *Hoekstra et al., 2016*; *Itsara et al., 2014*; *Kennedy et al., 2013*; *Samstag et al., 2018*; *Zheng et al., 2006*). While those studies have increased our understanding of mtDNA mutagenesis, these conclusions are largely based on a small number of tissue types. In this more extensive dataset, we observed a mean of 50 G→T/C→A mutations per sample with no samples having zero instances. In contrast, Arbeithuber et al. reported a mean of three G→T/C→A mutations per sample with ~17% of samples failing to detect this mutation type at all (*Arbeithuber et al., 2020*). Such low numbers can lead to significant biases in determining mutation frequencies and make it difficult to detect meaningful differences between sample types.

The age-related spectrum of mutations between tissues revealed considerable variation of the canonical ROS-associated G→T/C→A and G→C/C→G transversions (*Figure 2*). It was unexpected to find that such a significant proportion of mutations in some young tissues were ROS-linked, including 65% of all SNV mutations in the heart and 43% in skeletal muscle. Despite this, there was no increase of these mutations with age such that the proportion of ROS-linked mutation burden relative to all SNV mutations dropped to 30% and 25% heart and skeletal muscle, respectively. This finding suggests the possibility that different tissues experience varying levels of ROS injury in early life, but that these differences are kept in check during the aging process. Alternatively, it could mean that there is tissue specificity in how cells repair and/or destroy oxidatively damaged mitochondria and mtDNA, resulting in a steady state of ROS-linked mutations.

Interestingly, and consistent with prior reports, mouse tissues that are part of the CNS exhibited distinctly reduced levels of ROS-associated transversions compared to the other tissues we surveyed (*Figure 2*; *Hoekstra et al., 2016*; *Kennedy et al., 2013*; *Pickrell et al., 2015*; *Williams et al., 2013*). This suggests that studies focused only on brain tissue may under-represent the prevalence of ROS-linked somatic mtDNA mutations. Neural tissues are widely considered to be exquisitely sensitive to ETC dysfunction and therefore may have evolved a more robust ability to defend against it. More active repair or quality control mechanisms may help eliminate damaged mtDNA before inducing mutagenesis. Consistent with this idea, we observed that both heart and skeletal muscle have a high relative burden of G→T/C→A and G→C/C→G transversions, especially in young tissues, suggesting that their high metabolic activity may confer transversion mutations in these tissue types relative to transitions. These observations suggest that the previously broad conclusions of ROS being irrelevant in mtDNA mutagenesis may be biased for having relied on CNS tissues.

Our results with Elam and NMN intervention demonstrate that maintenance of mutations may be influenced by the origin of the mtDNA lesion. These pharmacological interventions are known to reverse age-related decline in mitochondrial function by improving the function of the mitochondrial pool in some aged tissues, albeit through very different mechanisms. Because neither drug has been shown to interact directly with mtDNA, the reduction of ROS-linked mtDNA mutation frequency within such a short treatment window suggests a loss of mtDNA harboring ROS-linked mutations, rather than a reduced rate of accumulation due to reduced excess ROS. We propose that rather than the incidence and impact of ROS damage on mtDNA being minimal, the direct or indirect recognition and removal of ROS-linked damage/mutations results in a steady state during aging. We expect that in tissues with exacerbated levels of ROS, the extent of damage could be sufficient to simultaneously

create multiple lesions along an individual genome. This could affect the formation of ROS-linked mutations if enough damage is accrued to a mitochondrion to target it for mitophagic removal, induce cellular apoptosis, or stall progression of the mtDNA replication fork. Any of these scenarios could effectively prevent ROS-induced mtDNA damage from being converted into a mutation or remove cells containing damaged mtDNA from the tissue, thereby shortening the detection window for mutations.

## Heteroplasmy vs. clonal expansion in aging

mtDNA mutations can be present as a fraction of mtDNA molecules within a cell (i.e. heteroplasmy) to all or nearly all molecules (i.e. homoplasmy). Our data demonstrate significant variability in mutational burden, as shown by the apparent heteroplasmic VAF, with kidney and liver exhibiting the highest loads (~10% and ~5%, respectively) in the tissues we sampled. However, despite an ever-increasing association with aging, disease, and injury, as well as being tractable with intervention, a common argument against the impact of mtDNA mutations in aging is that the observed heteroplasmy is well below the estimated at ~60–90% threshold has prevailed (*Rossignol et al., 2003*; *Rossignol et al., 1999*). However, this estimation is based on bulk level analysis of tissues. At the center of this issue is the inability of sequencing methods to distinguish between a low level of heteroplasmy arising from a number of mutations broadly distributed across a large area versus a high level of heteroplasmy found concentrated in a smaller population of cells. In the first case, the effect on cellular and tissue homeostasis would be minimal, but in the latter case, an effect on cellular function would be expected. For example, one could imagine a scenario where every cell contains a different homoplasmic pathogenic mutation which would negatively affect cellular function for every cell. In this case, mutations would be highly prevalent (100% of cells), yet individually rare. However, bulk sequencing would give the appearance that no mutation comes close to exceeding the phenotypic threshold. This issue is likely to be even more complicated in tissues that are highly heterogeneous in their cell-type composition, such as kidney and brain. Our data are silent as to the intra-cellular heteroplasmic level, cell distribution, and functional impact on tissues during aging. However, the broad variability in SNV mutation rates and detection of apparent heteroplasmic clones strongly suggests any impact is highly tissue-specific. Recent advances in single-cell sequencing will likely help tease out the tissue-specific impact of mtDNA mutations and are an important avenue to understand their role to the aging process (*Guo et al., 2023*; *Lareau et al., 2021*; *Sanchez-Contreras and Kennedy, 2022*).

## The 'clock-like' accumulation of mutations as a possible biomarker of aging

While the relative mutation frequencies in our dataset differed widely between aged tissues, we observed a surprisingly consistent twofold increase from young to old mice, driven largely by the accumulation of G→A/C→T transitions. Moreover, by combining our Duplex-Seq data with those from Arbeitheruber et al. our data show a clear clock-like behavior of these mutations (*Figure 2*), reminiscent of the Horvath epigenetic clock (*Horvath, 2013*). Such a clock-like accumulation holds promise as a biomarker for either biological or chronological age, depending on its modifiability. A key to the utility as a biomarker likely depends on the mechanistic driver(s) of this age-dependent accumulation. Specifically, mitochondrial G→A/C→T and, to a lesser extent, A→G/T→C transitions, are widely thought to arise either by deamination of cytidine or adenosine, respectively, or base misincorporations by Pol-γ (*Ameur et al., 2011*; *Kennedy et al., 2013*). We have previously used a subset of this dataset to demonstrate that a discontinuous strand-specific gradient of transitions exists across the mtDNA, consistent with deamination events arising from a long-lived single-stranded replication intermediate (*Sanchez-Contreras et al., 2021*). The slope of this gradient increases with age, indicating that deamination is likely an active chemical process (*Mikhailova et al., 2022*; *Sanchez-Contreras et al., 2021*). However, the underlying chemical mechanism behind these likely deaminations remains poorly understood and would impact its utility as a biomarker. For example, if deaminations are due to spontaneous hydrolysis, then transitions and/or the gradient is expected to be largely resistant to aging interventions and could serve as an indicator of chronological age. Alternatively, ROS and temperature, both affected by mitochondrial function, are known to induce deamination (*Chrétien et al., 2018*; *Frederico et al., 1990*; *Gates, 2009*; *Harman, 1972*). Therefore, interventions that affect

mitochondrial function and/or ROS production may be reflected in the rate of age-dependent accumulation of mtDNA transitions and thus act as an indicator of biological age.

In addition to the mechanism behind de novo mutations, cellular proliferation may play a role in the measured prevalence of age-dependent mutations. Our dataset indicates that of the tissues we examined, those with the highest proliferative index, namely kidney and liver, also exhibit the highest prevalence of mutations, as well as their higher percentage of clonal mtDNA heteroplasmy. This likely demonstrates the contribution of cellular proliferation to the presence of mtDNA mutation, regardless of their etiology. Specifically, as tissues proliferate, mutations that arise earlier will expand over time and have a higher chance of being detected. Therefore, interventions or environmental exposures that affect cellular proliferation would predict significant changes in age-dependent mutation prevalence. Clonal expansions of mtDNA mutations have been well documented in colonic tissues with age and are enhanced in dysplasia, indicating technical feasibility of using the clonal expansion of mtDNA mutations as a biomarker in at least some tissues (*Baker et al., 2019*; *Greaves et al., 2014*; *Greaves et al., 2006*).

## Alternative explanations for the results

We note two potential confounders that could provide an explanation for our data: nuclear-encoded mtDNA sequences (NUMTS) and shearing-induced artifacts arising from library construction. Studies indicate that NUMTS are highly prevalent and polymorphic in humans and is likely to be true in mice (*Calabrese et al., 2012*; *Richly and Leister, 2004*). Unknown and variations of NUMTS would be a potentially strong confounder in a study of outbred populations, but the use of one isogenic inbred line for this study likely eliminates this confounder. Specifically, this strong population bottleneck during colony formation would result in NUMTS-derived sequences that are shared across all the samples, especially between tissues from the same animal. However, we did take precautions to remove reads likely derived from NUMTS. First, we used the mm10 reference genome which is based on the C57BL/6J strain, so any NUMTS-derived variants present in our mtDNA data should preferentially align against any NUMTS. Second, we perform a BLAST on all reads containing at least one variant against the mm10 reference genome. We then reassigned the read based to whatever genomic location had the lower e-score. Lastly, we marked and remove variants shared between all individual samples. The result was an average of a dozen reads were removed, demonstrating that NUMTS are not likely a major source of false mutations.

The lack of age-associated increase in ROS-linked mutations makes it tempting to conclude that these mutations are either sequencing artifacts or at least inconsequential to the biology of aging. Countering these assumptions are two aspects of this study. First, treatment with FPG (*Figure 3—figure supplement 3*) and the analysis of SSCS variant (*Figure 3—figure supplement 2*) demonstrate that detected G→T/C→A and G→C/C→G mutation burdens are unlikely to be induced by damage to shearing-induced single-stranded regions (*Abascal et al., 2021*; *Xiong et al., 2022*). Second, the results demonstrate that ROS-linked mtDNA mutations can be decreased pharmacologically at late age and within a short treatment period in some tissues. Such an effect is difficult to reconcile with a signal resulting from manipulation of the DNA during library construction, which should affect all experimental groups approximately equally. We note there remains the possibility that if ROS-linked mutations were artifactual, but arose from preexisting biologically induced DNA damage, then changes in ROS production across different tissues or with interventions could result in apparent decline in ROS-linked mutations. Importantly, such a scenario would suggest biological relevancy of preventing damage to the mtDNA that ultimately keeps G→T/C→A and G→C/C→G mutations extremely low.

## Concluding remarks

Our data indicate that the accumulation of somatic mtDNA mutation is highly segmental in nature and that its impact on aging and/or disease risk may be tissue-dependent. Moreover, several of the tissues we examined, such as kidney, are very heterogeneous in their cell-type composition. By examining two compartments of brain (hippocampus and cerebellum), two compartments of the eye (retina and RPE/choroid), and two types of striated muscle tissue (skeletal and heart), we observe that not only are aging-related mtDNA mutation patterns tissue-specific, but they are also region- and cell type-specific. The different mutation spectra that we detected must be underlain by cell-specific regulation

of mtDNA. Previous overarching assumptions of how mtDNA mutations do, or do not, contribute to aging may have been premature when based on limited information from few tissues and with previous technical hurdles of poor accuracy in detection of low-level mutation burdens. Left unexplored in this study is whether an earlier initiation, longer duration, or more optimal dose of intervention would more robustly alter the mutation spectra, especially in other tissues that do not show an effect in this study. Additionally, we cannot discount that the lack of efficacy in some tissues may be due to tissue-specific differences in biodistribution of the two compounds tested. Finally, we did not have the opportunity or sufficient sample size in this study to explore potential correlation between the heterogeneity in mitochondrial function of tissues vs. the degree of change in the oxidative mutations. To fully understand how somatic mutation of mtDNA contributes to common diseases of aging, future work must delve into cell-specific mechanisms of mitochondrial mutation accumulation and mtDNA regulation, combined with high-accuracy methods of mtDNA mutation detection.

## Materials and methods
### Animals and tissue collection
C57BL/6J male mice from the National Institute of Aging Rodent Resource were handled according to the guidelines of the Institutional Animal Care Committee at the University of Washington. Two age cohorts were used at 4.5 and 26 months of age, respectively. Tissues from the 26-month-old cohort, including aged mice treated with Elam or NMN, were obtained from the same previously reported study, as previously described (*Whitson et al., 2020*). Briefly, 24-month-old mice were randomly assigned to control, Elam, or NMN treatment groups. Elam was provided by Stealth BioTherapeutics (Newton, MA, USA) and administered at a 3 mg/kg body weight/day dosage for 8 weeks through subcutaneously implanted osmotic minipumps (ALZET, Cupertino, CA, USA). Control mice were simultaneously housed in cages with Elam pump mice. NMN was obtained from the Imai Laboratory (Washington University in St Louis, MO, USA) and administered through ad libitum drinking water with a concentration based on each cage's measured water consumption and mean mouse body weight to approximate a 300 mg/kg/day dose. This method of drug delivery necessitated that NMN-treated mice were housed independently from control animals, however, treatments were run concurrently. Because we sequenced just a subset of the animals from the Whitson et al. study (N=3–5 vs. N=11–15 each group), we minimized study variation by excluding tissues from animals with clearly cancerous lesions by gross analysis (primarily seen in liver) and then sequenced a random cohort of age-matched groups from the three treatment cohorts (*Whitson et al., 2020*).

Mouse tissue was collected immediately following euthanasia. Representative portions of six different organ systems were flash-frozen: (1) apex of the heart; (2) 2 mm section from the inferior pole of the left lateral liver lobe; (3) eyes were enucleated and cleared of muscle and adipose tissue before dissecting the retina from the RPE-choroid complex (also referred to as 'eye cup' or 'EC' in our raw data files) with both regions preserved separately; (4) 3 mm slice of the lower pole of the decapsulated left kidney; (5) proximal 3 mm of left gastrocnemius; (6) brain was dissected in ice-cold 1× PBS to obtain a 3-mm-thick coronal section from the most anterior/septal pole of the left hippocampus and a 3-mm-thick sagittal section from the medial side of the left cerebellar hemisphere. For every sample, dissecting tools were wiped in 70% ethanol, a new razor blades and cutting boards were used, and samples were rinsed in fresh 1× PBS to minimize the contribution of blood and avoid DNA cross-contamination. For perfused experiments, a separate cohort of NIA male mice matching the same age for the aged cohort above (26 months, N=3) was perfused transcardially with 1× PBS containing calcium and magnesium before tissue isolation.

### DNA processing and Duplex-Seq
DNA was extracted using the Qiagen DNeasy Blood and Tissue kit (Qiagen, Valencia, CA, USA) and stored at –80°C. Duplex-Seq was performed as previously described (*Kennedy et al., 2014*), but with several modifications also previously described (*Hoekstra et al., 2016*; *Sanchez-Contreras et al., 2021*). Duplex-Seq adapters with defined unique molecular identifiers (UMIs) were used and were constructed by separately annealing complementary oligonucleotides (IDT, Coralville, IA, USA), each containing 1 of 96 UMIs of defined sequence (*Supplementary file 1*—Table 1). The resulting adapters were diluted to 25 µM for ligation to sheared DNA. Targeted capture used the IDT xGen Lockdown

protocol and probes specific for mouse mtDNA (Integrated DNA Technologies, Coralville, IA, USA) following the manufacturer's instructions (*Supplementary file 1*—Table 2). The resulting libraries were indexed and then sequenced using ~150-cycle paired-end reads (300 cycles total) on an Illumina NovaSeq6000 with ~$20 \times 10^6$ reads per sample. Per sample sequencing metrics are available in *Supplementary file 1*—Table 3.

## mtDNA content

mtDNA copy number was determined by droplet-digital PCR (ddPCR) by the Genomic Sciences Core of the Oklahoma Nathan Shock Center. Briefly, 200 ng of total genomic DNA from the same isolated DNA sample used for Duplex-Seq was mixed with ddPCR assay components including fluorogenic 'TaqMan' primer probe sets and the reactions were distributed across a chip with ~20,000 ~ 1 nL droplets to dilute the DNA template to either zero or one copy per well, as described previously (*Masser et al., 2016*). Reactions were then cycled to end-point and fluorescence was read in each droplet. Based on the count of fluorescent positive and negative wells and using a Poisson distribution, the number of target copies was calculated per microliter. nDNA counting was performed in parallel and used as a surrogate for cell number which allows for normalization to cell number and results in an absolute quantitation of mtDNA. The data are available in *Supplementary file 1*—Table 4.

## Data analysis and statistics

The raw sequencing data was processed using version 1.1.4 of our in-house bioinformatics pipeline (https://github.com/Kennedy-Lab-UW/Duplex-Seq-Pipeline) with the default consensus making parameters. A detailed description of the Duplex-Seq pipeline is described in *Sanchez-Contreras et al., 2021*. Seven small polynucleotide repeats were masked to reduce errors associated with alignment artifacts (*Supplementary file 1*—Table 5). To reduce the potential impact of NUMTS, we subjected all reads containing non-homoplasmic variants to BLAST-based alignment against a database containing potential common contaminants (dog: canFam3; bovine: bosTau9; nematode: ce11; mouse: mm10; human: hg38; rat: Rnor 6.0). The inclusion of the mm10 genome in this database also allows for the identification of pseudogenes due to BLAST's high sensitivity. Reads where the alignment with the lowest e-score is the same as the original alignment are kept and the remaining reads and associated variants are removed from further analysis. In addition, we removed variants found across all samples within an individual mouse, which would indicate an inherited, and not a somatic, process.

To quantify the frequency of de novo events, we used a clonality cutoff of >1% or a depth of <100, which excluded any positions with variants occurring at a high heteroplasmy level and scores each type of mutation only once at each genome position. Called variants were annotated using the Ensembl Variant Effect Predictor v99 to obtain protein change. Mutation frequencies were calculated by dividing the number of reads for each allele by the total number of reads at the same mtDNA position. Correlation statistics were applied to determine intra- and inter-animal mutation frequency variation using GraphPad Prism, R, and Python software. Statistical significances between young and old mutation frequencies for SNVs and In/Dels were determined by Welch's t-test for each mutation class between young vs. old. For comparisons within each age group and mutation type but between tissues (e.g. young kidney vs. young liver vs. young heart frequency, etc.), one-way repeated measures (i.e. each tissue) ANOVA were followed with Tukey's HSD. For comparing more than one group to a control (e.g. frequency of mutation type for kidney in aged ELAM or aged NMN treated mice vs. untreated aged control) one-way ANOVA was followed by Dunnett's multiple comparison test. The significance of the ratio of means between young and old mutation spectra was determined by t-test for the ratio of means of two independent samples from two Gaussian distributions with the 95% confidence interval estimated by Fieller's theorem implemented in the *mratios* R-package (*Fieller, 1954*). p-Values or adjusted p-values ($p_{adj}$) less than 0.05 were considered significant in all cases.

Somatic heteroplasmic clones were defined as variants called in >2 supporting reads. Expected clone events were calculated as the percentage of clonal mutations (>2 calls per sample) for each mutation type observed by age, either in total across all samples or by individual tissue types (as indicated). Significance between expected and observed clone events was calculated by Poisson distribution.

dN/dS analysis was carried out using the *dNdScv* R package (*Martincorena et al., 2017*). The package is designed to quantify selection in somatic evolution by implementing maximum-likelihood methods that accounts for trinucleotide context-dependent substitution models, which is highly biased in mtDNA. Each sample was processed independently using the *dNdScv* implementation with options *max_muts_per_gene_per_sample* set to *Inf*, *numcode* set to 2, and the mean depth per gene included as a covariate. ND6 was analyzed separately due to it residing on the opposite strand from the other protein coding genes and having different G/C-skew (*Ju et al., 2014*). The resulting dN/dS values ($w_{mis}$) for each gene were averaged and significant deviation from 1 determined by a one sample t-test with Bonferroni correction (significance set to $p \le 0.0167$).

## Data availability and reproducibility

The Duplex-Seq-Pipeline is written in Python and R, but has dependencies written in other languages. The DuplexSeq-Pipeline software has been tested to run on Linux, Windows WSL1, Windows WSL2, and Apple OSX. The software can be obtained at . Raw mouse sequencing data used in this study are available at SRA accession PRJNA727407. The data from Arbeithuber et al. are available at SRA accession PRJNA563921. The final post-processed data, including variant call files, depth information, data summaries, and mutation frequencies, as well as the scripts to perform reproducible production of statistics and figure generation (with the exception of *Figure 5C–E*) are available at https://github.com/Kennedy-Lab-UW/Sanchez_Contreras_etal_2023.

## Acknowledgements

The authors wish to acknowledge the technical support of Carolyn N Mann for the Sweetwyne Laboratory and Dr Claudia Moreno for the helpful discussions and long sessions of scientific writing.

NIA P01 AG001751 (PSR and DM), NIA K01 AG062757 (MTS), NIDDK R21 DK128540 (MTS and MS-C), Oklahoma Nathan Shock Center Pilot Award (MS-C and MTS), Dolsen Family Gift Funds, DOD/CDMRP W81XWH-16-1-0579, NHGRI R21 HG011229, and NCI R21 CA259780 (SRK), Genetic Approaches to Aging Training Grant NIA T32 AG000057 (JW and KAT), Biological Mechanisms for Healthy Aging Training Grant NIA T32 AG066574 (KAT).

## Additional information

### Competing interests

Scott R Kennedy: is an equity holder and paid consultant for Twinstrand Biosciences, a for-profit company commercializing Duplex Sequencing. No Twinstrand products were used in the generation of the data. The other authors declare that no competing interests exist.

### Funding

| Funder | Grant reference number | Author |
|---|---|---|
| National Institute on Aging | P01AG001751 | David J Marcinek Peter S Rabinovitch |
| National Institute of Diabetes and Digestive and Kidney Diseases | R21DK128540 | Monica Sanchez-Contreras |
| Congressionally Directed Medical Research Programs | W81XWH-16-1-0579 | Scott R Kennedy |
| National Human Genome Research Institute | R21HG011229 | Scott R Kennedy |
| National Cancer Institute | R21CA259780 | Scott R Kennedy |
| National Institute on Aging | K01AG062757 | Mariya T Sweetwyne |

| Funder | Grant reference number | Author |
|---|---|---|
| National Institute on Aging | T32AG000057 | Kristine A Tsantilas<br>Jeremy A Whitson |
| National Institute on Aging | T32AG066574 | Kristine A Tsantilas |

The funders had no role in study design, data collection and interpretation, or the decision to submit the work for publication.

## Author contributions

Monica Sanchez-Contreras, Conceptualization, Formal analysis, Investigation, Visualization, Writing – original draft, Writing – review and editing; Mariya T Sweetwyne, Conceptualization, Formal analysis, Supervision, Investigation, Visualization, Writing – original draft, Writing – review and editing; Kristine A Tsantilas, Jeremy A Whitson, Matthew D Campbell, Resources, Investigation; Brenden F Kohrn, Software; Hyeon Jeong Kim, Formal analysis; Michael J Hipp, Jeanne Fredrickson, Megan M Nguyen, Investigation; James B Hurley, Resources; David J Marcinek, Peter S Rabinovitch, Funding acquisition, Writing – review and editing; Scott R Kennedy, Data curation, Software, Formal analysis, Supervision, Funding acquisition, Visualization, Methodology, Writing – original draft, Writing – review and editing

## Author ORCIDs

Monica Sanchez-Contreras http://orcid.org/0000-0002-3092-2781
Kristine A Tsantilas http://orcid.org/0000-0002-4274-6930
Brenden F Kohrn http://orcid.org/0000-0001-9948-2131
Michael J Hipp http://orcid.org/0000-0003-1904-0670
James B Hurley http://orcid.org/0000-0002-7754-0705
Peter S Rabinovitch http://orcid.org/0000-0001-7169-3543
Scott R Kennedy http://orcid.org/0000-0002-4444-1145

## Ethics

This study was performed in accordance to the Guide for the Care and Use of Laboratory Animals of the National Institutes of Health. All of the animals were handled according to an approved institutional animal care and use committee (IACUC) protocol (2174-23) at the University of Washington.

## Decision letter and Author response

Decision letter https://doi.org/10.7554/eLife.83395.sa1
Author response https://doi.org/10.7554/eLife.83395.sa2

## Additional files

### Supplementary files

• Supplementary file 1. File containing Supplementary Tables 1-5. Table 1. Sequence of the 96 defined unique molecular identifier (UMI) duplex sequencing adapters. Sequence is provided in 5'→3' orientation. Complementary UMI sequences are highlighted in red. Table 2. Sequence of the mouse-specific capture probes against the mtDNA. Sequence is provided 5'→3' orientation. Biotin moiety is denoted by '/5Biosg/'. Table 3. Summary of duplex sequencing data. Summary of the samples sequenced, including assay performance metrics, including mitochondrial genome (mtDNA) enrichment specificity, family size, and consensus metrics, bases sequenced, sequencing depth, mutation counts, and mutation frequencies. Table 4. Mitochondrial genome (mtDNA) to nuclear genome (nDNA) copy number ratio data. Summary of the samples used to determine mtDNA and nDNA copy numbers. ND = not determined. Table 5. Genome coordinates of regions masked in the analysis. Coordinates are 1-indexed and columns are in .bed format order.

• Supplementary file 2. List of all detected mutations.

• MDAR checklist

### Data availability

The Duplex-Seq-Pipeline is written in Python and R, but has dependencies written in other languages. The DuplexSeq-Pipeline software has been tested to run on Linux, Windows WSL1, Windows WSL2 and Apple OSX. The software can be obtained at https://github.com/Kennedy-Lab-UW/Duplex-Seq-Pipeline, (copy archived at swh:1:rev:878457d08f3272db01550fbf85da631ae11b713c). Raw mouse

sequencing data used in this study are available at SRA accession PRJNA727407. The data from Arbeithuber et al. are available at SRA accession PRJNA563921. The final post-processed data, including variant call files, depth information, data summaries, and mutation frequencies, as well as the scripts to perform reproducible production of statistics and figure generation (with the exception of Figure 5C-E) are available at https://github.com/Kennedy-Lab-UW/Sanchez_Contreras_etal_2022, (copy archived at swh:1:rev:ff5f6a2bf4b30f40fd80ad4136ceea58f1b2dd52).

The following dataset was generated:

| Author(s) | Year | Dataset title | Dataset URL | Database and Identifier |
|---|---|---|---|---|
| Sanchez-Conteras M, Sweetwyne MT, Kennedy SR | 2021 | Mitochondrial DNA isolated from different tissues with two different ages (4-5 months and 24-26 months) and two different aging interventions | https://www.ncbi.nlm.nih.gov/bioproject/?term=PRJNA727407 | NCBI BioProject, PRJNA727407 |

The following previously published dataset was used:

| Author(s) | Year | Dataset title | Dataset URL | Database and Identifier |
|---|---|---|---|---|
| Arbeithuber B, Makova KD | 2019 | Duplex sequencing of mtDNA from mouse somatic tissues (brain and skeletal muscle), single oocytes, and oocyte pools was performed to study the effect of aging on mtDNA substitution frequencies | https://www.ncbi.nlm.nih.gov/bioproject/?term=PRJNA563921 | NCBI BioProject, PRJNA563921 |

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
