## [Editor Report]

Using the most accurate deep sequencing technology, duplex sequencing, these authors have detected over 89,000 independent somatic mtDNA mutations representing the largest catalog of somatic mtDNA point mutations during aging in a single study. The analysis of these mutations provides compelling evidence to dismiss the idea that reactive oxygen species are a driver of mtDNA mutagenesis, but suggests that ROS may be tissue dependent. These results should provide a fundamental understanding of mitochondrial DNA mutagenesis in aging that should appeal to a broad audience. The novel discovery is the significant presence of transversion mutations (C>A/G>T and C>G/G>C), which previously were assumed almost nonexistent. Moreover, the study finds that, unlike conventional mtDNA mutations, these transversions are not involved in clonal expansion and do not accumulate with age; their relative presence varies very significantly between tissues and can be affected by drug interventions.

---

## [Decision Letter]

**Decision letter after peer review:**

Thank you for submitting your article "The multi-tissue landscape of somatic mtDNA mutations indicates tissue-specific accumulation and removal in aging" for consideration by *eLife*. Your article has been reviewed by 2 peer reviewers, one of whom is a member of our Board of Reviewing Editors, and the evaluation has been overseen by Molly Przeworski as the Senior Editor. The following individual involved in the review of your submission has agreed to reveal their identity: Konstantin Khrapko (Reviewer #2).

Please address these essential items in your revised manuscript.

1. Please clarify that your claim of the largest collection of mtDNA mutations in a single study is limited to 'point' mutations only. An early study of mtDNA deletions detected much more mtDNA alterations.

2. Please address reviewer 1's concerns about how DNA mutations in NUMTs are excluded from your study.

3. Reviewer 1 has a concern that the effect of the drug treatment is independent of point mutagenesis for ROS-derived mutations. Given that the low amount of ROS-linked mutations is already well below a physiological level that would cause loss of mitochondrial function, please explain how reducing these mutations improves mitochondrial function.

4. Both of the reviewers agree that your study finds that, unlike conventional mtDNA mutations, the transversions are not involved in clonal expansion and do not accumulate with age and their relative presence varies very significantly between tissues and is affected by drug interventions. This unusual behavior of transversions is difficult to explain and thus this study is thought-provoking and will likely lead to further developments in mtDNA field. However, Reviewer 2 is concerned that the detection of transversions, in particular, may be due to an artifact reported by Abascal 2021. Please address their comments.

*Reviewer #1 (Recommendations for the authors):*

p.9. What is the sensitivity of In/Del detection compared to pt mutations?

"255 determined the frequency of both SNVs and small InDels (≲15bp) in aged mice."

Please reiterate this indel size range in the Discussion.

p. 16:

"414 Late-life treatment with mitochondrially-targeted interventions eliminate ROS-linked mutations".

p. 17:

"(Control: 9.7{plus minus}3.0x10-7 vs. ELAM: 5.5{plus minus}1.5x10-7, p=0.019; Control: 9.7{plus minus}3.0x10-7 vs. NMN: 440 4.7{plus minus}0.9x10-7, p=0.017)"

Beware of multiple hypothesis testing. In the Methods, they state that they accept tests with p-values < 0.05. If they are testing 6 pairs of substitution types and 2 treatments, then for a family-wise error rate of 0.05 their cutoff should really be p < 0.00427 (Šidák correction assuming 6x2=12 tests).

p. 21:

"This last scenario is a blind spot in Duplex-Seq methodology, such that large deletions are difficult to detect due to the reliance on the alignment of DNA fragments of ~200-400 nucleotides."

Studies exist that fill this blind spot. Please reference them, especially in light of the unreferenced assertion in the previous sentence.

"Our study is the first to demonstrate that ROS-linked mtDNA mutations can be specifically decreased pharmacologically at a late age and within a short treatment period in some tissues."

This assertion requires acceptance of the very marginal p-values on page 17.

p. 22:

"Previous overarching assumptions of how mtDNA mutations do, or do not, contribute to aging may have been premature when based on limited information from few tissues and with previous technical hurdles of poor accuracy in detection of low-level mutation burdens."

Wholeheartedly agreed. Do you think that the rates found herein are high enough to cause any of the phenotypes of aging? Would you care to defend your answer either way in the manuscript?

*Reviewer #2 (Recommendations for the authors):*

Historical comment:

Previous studies in the field are described in the following paragraph: line 503 "The findings reported here broadly recapitulate those from smaller studies reporting that somatic mtDNA point mutations occur at a frequency on the order of 10-6, increase with age, and are biased towards G→A/C→T transitions (Ameur et al., 2011; Andreazza et al., 2019; Arbeithuber et al., 2020; Hoekstra et al., 2016; Kennedy et al., 2013; Samstag et al., 2018; Williams et al., 2013)"

We note, for the sake of fairness, that back in 1997 we published a study on somatic mtDNA mutations in human tissues (Khrapko et al. PNAS Vol. 94, pp. 13798-13803). In this work, we established the correct ballpark of mtDNA mutational frequency for SNVs (~3x10-6), demonstrated the strong bias towards G→A/C→T transitions, and, additionally, inferred the existence of clonal expansions/clusters of somatic mtDNA mutations within tissues. Later we also established the extreme strand asymmetry of somatic mtDNA mutations – Zheng 2006 – (doi:10.1016/j.mrfmmm.2005.12.012).

Interestingly, one of the controls we used to validate our mutational spectra was based on an asymmetric PCR test, in which we separately amplified either Watson or Crick strand. We then compared the Watson and the Crick mutational spectra to detect and exclude any single-stranded non-mutational base changes, i.e. those present in one but not the other spectrum (Khrapko et al., Nucleic Acids Research, 1997, Vol. 25, No. 4 685-693). This approach is conceptually very similar to the ds sequencing. It would be nice to cite and discuss this 'prior art research' in this publication.

Additionally, I would also recommend making other data more accessible to the reader, e.g., publishing the full list of mutations (and half-mutations) as a simple downloadable excel sheet.

---

## [Author Response]

Reviewer #1 (Recommendations for the authors):p.9. What is the sensitivity of In/Del detection compared to pt mutations?

We can’t provide a specific ratio for this question, but Duplex-Seq is undoubtedly better at detecting point mutations. This is for several reasons. First, we perform targeted DNA capture using probes against the “wild-type” mtDNA sequence and the more sequence that is missing or added by an In/Del, the less efficient the capture will be. For small In/Dels, there is likely little loss, but for larger structural variations it’s likely to be a major source of loss. Second, the main aligner we use, *bwa*, is not very sensitive to larger In/Dels. An alternative aligner, of which there are many, would need to be used to get at this information. As the reviewer pointed out, Lujan et al. have developed a nice computational method to detect large structural variants from short read technologies. As we perform a reference free consensus calling algorithm any structural variations present in the samples will be found in the generated fastq files, provided that they were captured during target enrichment.

"255 determined the frequency of both SNVs and small InDels (≲15bp) in aged mice."Please reiterate this indel size range in the Discussion.

We have done so.

p. 16:"414 Late-Life treatment with mitochondrially-targeted interventions eliminate ROS-linked mutations".

There doesn’t seem to be a specific comment here, but we have changed “eliminates” to “reduces” to better reflect the fact that ROS-linked mutations are still present.

p. 17:"(Control: 9.7{plus minus}3.0x10-7 vs. ELAM: 5.5{plus minus}1.5x10-7, p=0.019; Control: 9.7{plus minus}3.0x10-7 vs. NMN: 440 4.7{plus minus}0.9x10-7, p=0.017)"Beware of multiple hypothesis testing. In the Methods, they state that they accept tests with p-values < 0.05. If they are testing 6 pairs of substitution types and 2 treatments, then for a family-wise error rate of 0.05 their cutoff should really be p < 0.00427 (Šidák correction assuming 6x2=12 tests).

We apologize for the confusion on this. The stated p-values are in fact adjusted p-values and we do perform multiple testing correction (specifically Tukey’s HSD or Dunnet’s test, where applicable). We now refer to the multiple testing corrected p-value as p_adj_.

p. 21:"This last scenario is a blind spot in Duplex-Seq methodology, such that large deletions are difficult to detect due to the reliance on the alignment of DNA fragments of ~200-400 nucleotides."Studies exist that fill this blind spot. Please reference them, especially in light of the unreferenced assertion in the previous sentence.

We have revised this statement slightly and have also cited Krishnan et al., 2008 and Lujan et al., 2012 in the preceding sentence.

"Our study is the first to demonstrate that ROS-linked mtDNA mutations can be specifically decreased pharmacologically at a late age and within a short treatment period in some tissues."This assertion requires acceptance of the very marginal p-values on page 17.

We have toned down this assertion to read

“Our study suggests that mtDNA mutations can be specifically decreased pharmacologically at late age and within a short treatment period in some tissues.”

And

“This response provides a potential explanation for why ROS-linked mtDNA mutations are rarely found to accumulate with age.”

See also our reply to the reviewer’s concern regarding p-values in comment #9.

p. 22:"Previous overarching assumptions of how mtDNA mutations do, or do not, contribute to aging may have been premature when based on limited information from few tissues and with previous technical hurdles of poor accuracy in detection of low-level mutation burdens."Wholeheartedly agreed. Do you think that the rates found herein are high enough to cause any of the phenotypes of aging? Would you care to defend your answer either way in the manuscript?

As noted in Reply #3, we need to know both how many cells harbor a mutation, as well as what the heteroplasmic level is within the cell, before making claims on their pathological impact. We have included a paragraph in the Discussion about this important issue starting on line 582.

Reviewer #2 (Recommendations for the authors):Historical comment:Previous studies in the field are described in the following paragraph: line 503 "The findings reported here broadly recapitulate those from smaller studies reporting that somatic mtDNA point mutations occur at a frequency on the order of 10-6, increase with age, and are biased towards G→A/C→T transitions (Ameur et al., 2011; Andreazza et al., 2019; Arbeithuber et al., 2020; Hoekstra et al., 2016; Kennedy et al., 2013; Samstag et al., 2018; Williams et al., 2013)"We note, for the sake of fairness, that back in 1997 we published a study on somatic mtDNA mutations in human tissues (Khrapko et al. PNAS Vol. 94, pp. 13798-13803). In this work, we established the correct ballpark of mtDNA mutational frequency for SNVs (~3x10-6), demonstrated the strong bias towards G→A/C→T transitions, and, additionally, inferred the existence of clonal expansions/clusters of somatic mtDNA mutations within tissues. Later we also established the extreme strand asymmetry of somatic mtDNA mutations – Zheng 2006 – (doi:10.1016/j.mrfmmm.2005.12.012).Interestingly, one of the controls we used to validate our mutational spectra was based on an asymmetric PCR test, in which we separately amplified either Watson or Crick strand. We then compared the Watson and the Crick mutational spectra to detect and exclude any single-stranded non-mutational base changes, i.e. those present in one but not the other spectrum (Khrapko et al., Nucleic Acids Research, 1997, Vol. 25, No. 4 685-693). This approach is conceptually very similar to the ds sequencing. It would be nice to cite and discuss this 'prior art research' in this publication.

We apologize for not including these citations. We are definitely aware of them and have cited them in prior publications. This was an oversight on our part, and we have now included them in the revised manuscript.

Additionally, I would also recommend making other data more accessible to the reader, e.g., publishing the full list of mutations (and half-mutations) as a simple downloadable excel sheet.

An excellent suggestion. It is available as Supplementary File 2.